# Comparison of clinical characteristics of Zika and dengue symptomatic infections and other acute illnesses of unidentified origin in Mexico

**Pablo F. Belaunzarán-Zamudio**[1¤a]*, **Allyson Mateja**[2], **Paola del Carmen Guerra-de-Blas**[3], **Héctor A. Rincón-León**[4], **Karla Navarro-Fuentes**[4], **Emilia Ruiz-Hernández**[5], **Sandra Caballero-Sosa**[6], **Francisco Camas-Durán**[6], **Zoila Priego-Smith**[6], **José G Nájera-Cancino**[7], **Alexander López-Roblero**[7¤b], **Karina del Carmen Trujillo-Murillo**[7¤c], **John H. Powers**[8], **Sally Hunsberger**[8], **Sophia Siddiqui**[8], **John H. Beigel**[8], **Raydel Valdés-Salgado**[9], **Guillermo Ruiz-Palacios**[1], **the Mexican Emerging Infectious Diseases Clinical Research Network (LaRed)**[¶]

**1** Departamento de Infectología, Instituto Nacional de Ciencias Médicas y Nutrición Salvador Zubirán, Mexico City, Mexico, **2** Clinical Monitoring Research Program Directorate, Frederick National Laboratory for Cancer Research, Frederick, Maryland, United States of America, **3** The Mexican Emerging Infectious Diseases Clinical Research Network (LaRed), Mexico City, Mexico, **4** Unidad de Medicina Familiar No.11, Instituto Mexicano del Seguro Social, Tapachula, Chiapas, Mexico, **5** Hospital General de Tapachula, Tapachula, Chiapas, Mexico, **6** Clínica Hospital Dr. Roberto Nettel Flores, Instituto de Seguridad y Servicios Sociales de los Trabajadores del Estado, Tapachula, Chiapas, Mexico, **7** Hospital Regional de Alta Especialidad Ciudad Salud, Tapachula, Chiapas, Mexico, **8** National Institute of Allergy and Infectious Diseases, Bethesda, Maryland, United States of America, **9** Westat, Rockville, Maryland, United States of America

¤a  Current address: National Institute of Allergy and Infectious Diseases, Bethesda, Maryland, United States of America
¤b  Current address: Facultad de Ciencias Químicas, Campus IV, Universidad Autónoma de Chiapas, Chiapas, Mexico
¤c  Current address: Facultad de Medicina Humana "Dr. Manuel Velasco Suárez", Campus IV, Universidad Autónoma de Chiapas, Chiapas, Mexico
¶ Membership of The Mexican Emerging Infectious Diseases Clinical Research Network (LaRed) is provided in the Acknowledgments.
* p.belaunz@infecto.mx

## Abstract

### Background

Our purpose was to provide a detailed clinical description, of symptoms and laboratory abnormalities, and temporality in patients with confirmed Zika and dengue infections, and other acute illnesses of unidentified origin (AIUO).

### Methods/ Principal findings

This was a two-year, multicenter, observational, prospective, cohort study. We collected data from patients meeting the Pan American Health Organization's modified case-definition criteria for probable Zika infection. We identified Zika, dengue chikungunya by RT-PCR in serum and urine. We compared characteristics between patients with confirmed Zika and dengue infections, Zika and AIUO, and Dengue and AIUO at baseline, Days

**Data Availability Statement:** The datasets for this manuscript are not publicly available because it is needed a data-sharing agreement that provides for: (1) a commitment to using the data only for research purposes and not to identify any individual participant; (2) a commitment to securing the data using appropriate computer technology; and (3) a commitment to destroying or returning the data after analyses are completed. Requests to access the datasets should be directed to The Mexican Emerging Infectious Diseases Clinical Research Network (LaRed) (www.redmexei.mx; lared-cc@redmexei.mx).

**Funding:** This work was supported by The Mexican Emerging Infectious Diseases Clinical Research Network (LaRed). LaRed is funded by the Mexico Ministry of Health and the U.S. National Institute of Allergy and Infectious Diseases. This study was supported in part by Consejo Nacional de Ciencia y Tecnología [FONSEC SSA/IMSS/ISSSTE Projects No. 71260 and No. 127088]; National Institute of Allergy and Infectious Diseases, National Institutes of Health, through its Intramural Research Programs and a contract with Westat, Inc., Contract Number: HHSN27222009000031, Task Order Number: HHSN27200002; and in part with federal funds from the National Cancer Institute, National Institutes of Health, under Contract No. HHSN261200800001E and Contract No. 75N91019D00024, Task Order No. 75N91019F00130. The funders had no role in study design, data collection and analysis, decision to publish, or preparation of the manuscript.

**Competing interests:** The authors have declared that no competing interests exist.

3,7,28 and 180 of follow-up. Most episodes (67%) consistent with the PAHO definition of probable Zika could not be confirmed as due to any flavivirus and classified as Acute Illnesses of Unidentified Origin (AIUO). Infections by Zika and dengue accounted for 8.4% and 16% of episodes. Dengue patients presented with fever, generalized non-macular rash, arthralgia, and petechiae more frequently than patients with Zika during the first 10 days of symptoms. Dengue patients presented with more laboratory abnormalities (lower neutrophils, lymphocytosis, thrombocytopenia and abnormal liver function tests), with thrombocytopenia lasting for 28 days. Zika patients had conjunctivitis, photophobia and localized macular rash more frequently than others. Few differences persisted longer than 10 days after symptoms initiation: conjunctivitis in Zika infections, and self-reported rash and petechia in dengue infections.

## Conclusions

Our study helps characterize the variety and duration of clinical features in patients with Zika, dengue and AIUO. The lack of diagnosis in most patients points to need for better diagnostics to assist clinicians in making specific etiologic diagnoses.

## Author summary

Zika and dengue virus infections present a wide variety of symptoms that overlap with other acute illnesses. Our study helps characterize the variety and duration of clinical features in patients with Zika, dengue and other acute illnesses of unidentified origin (AIUO). We collected data from 441 patients seeking care for symptoms compatible with Zika infection based on a PAHO definition with onset in the previous 7 days in Tapachula, Mexico. We identified Zika, dengue, and chikungunya infections using an RT-PCR in serum and urine. We could not determine which pathogen caused the symptoms in most episodes (67%) and these were classified as AIUO. We observed differences in frequency and duration of clinical manifestations between patients with Zika, dengue and AIUO. Dengue tended to be a more symptomatic and disabling disease with generalized symptoms and more laboratory alterations that lasted longer than Zika and AIUO. Patients with Zika presented more frequently eye symptoms and localized rash. Nonetheless, we observed substantial overlap across diseases, and it remains unclear whether symptoms alone can distinguish these diseases in individual patients. The lack of diagnosis in most patients points to need for better diagnostics to assist clinicians in making specific etiologic diagnoses.

## Introduction

Zika virus infection presents a wide variety of clinical symptoms that may go from an asymptomatic infection to an influenza-like illness [1] or meningitis and encephalitis [2]. Patients in small studies with symptomatic Zika virus infection characteristically presented with rash, conjunctivitis, malaise, myalgia, arthralgia, edema, headache, retro-ocular pain and fever [3,4]. Less frequent manifestations were thrombocytopenia [5] and lymphadenopathies [6]. It is difficult to clinically differentiate between Zika, dengue and chikungunya virus infections, which are regionally endemic [7–9] given the overlap in symptoms. Moreover, concurrent outbreaks

and even co-infections appear to be relatively frequent [8,10,11]; and clinical presentation may vary geographically or over time [12]. Here, we describe the clinical characteristics and laboratory abnormalities in patients with confirmed Zika or dengue infections, and other acute illnesses of unidentified origin (AIUO) observed in a cohort of people with symptoms compatible with Zika infection [13] in four clinical sites in Tapachula in the Mexico-Guatemalan border. Our purpose is to provide a detailed clinical description of these infections according to their etiology, including temporality of symptoms.

## Methods

### Ethics statement

The study protocol was evaluated and approved by the Institutional Review Board of Instituto Nacional de Ciencias Médicas y Nutrición Salvador Zubirán in all Mexican participating institutions. Participation was voluntary and documented through a written informed consent procedure. Participants younger than 18 years were requested their assent and parents or legal tutors authorized their participation.

### Study design and settings

This study uses an observational, prospective, cohort study design to collect information on the natural history of Zika. The study was conducted between June 2016 and June 2018 in four participating health care centers (one primary, ambulatory healthcare center, two general hospitals and a third-level, referral hospital) in the city of Tapachula, Chiapas.

### Study population and definitions

We included patients that were 12 years of age or older; seeking care for acute fever and/or rash episodes and followed them up for 6 months. Subjects were enrolled in the cohort if they met a modified version of the criteria for probable Zika virus infection outlined by the Pan American Health Organization [14]. This comprises rash or elevated body temperature ($> 37.2 °C$) accompanied with at least one of either arthralgia, myalgia, non-purulent conjunctivitis or conjunctival hyperemia, headache or malaise in the 7 previous days before the initial visit, with no obvious alternative diagnosis to explain the symptoms. In this analysis, we describe the frequency of symptoms and signs, physical findings and clinical laboratory abnormalities at baseline and during the first 28 days of enrollment, and compare findings between patients with Zika, dengue and AIUO.

We defined confirmed Zika, chikungunya and dengue infections if participants had detectable RNA in blood or urine samples at baseline or at any timepoint during the first week of follow-up after the initial visit (scheduled at days 3 and 7). When RT-PCR was negative for these arboviruses in all samples, episodes were considered as an AIUO. Patients with no available RT-PCR tests results in urine and blood samples on two or more visits and negative RT-PCR test results were classified as missing end excluded from comparisons.

### Procedures

At baseline, we collected information on sociodemographic, symptoms, and performed a complete physical exam, including a detailed neurological exam. We assessed the impact of the disease using the World Health Organization Disability Assessment Schedule 2.0 (WHODAS 2). The WHODAS 2 is an instrument designed to provide a cross-cultural standardized method for measuring activity limitations and participation restrictions irrespective of the individual's medical diagnosis [15]. The instrument evaluates 6 domains of functioning: cognition,

mobility, self-care, interaction with other people, life activities, and participation; and provides a score to produce a standardized and comparable disability measure [15]. We repeated the assessments 3,7, 28 and 180 days after enrollment. In this report, we include data up to 28 days after enrollment. We collected blood and urine samples and performed complete blood count and blood chemistries and performed on the same sample RNA identification of Zika, dengue chikungunya and panflavivirus by Reverse Transcription Polymerase Chain Reaction in serum and urine as previously described [13]. As patients had started symptoms at any time in the 7 days before enrollment, we assessed patients within a range of overlapping length of symptoms at each visit: Baseline visit includes symptoms within the first 7 days of initiation; Day 3 visit up to 10 days of symptom initiation; Day 7 visit includes symptoms between days 10 to 14 after symptom initiation; and Day 28 visit up to 35 days after symptom initiation. Findings at physical exam are described if present at the corresponding visit.

## Statistical analysis

We used simple proportions and medians as measures of central tendency with the corresponding interquartile ranges for descriptive purposes. There were too few patients with chikungunya, so we compared characteristics between patients with confirmed Zika and dengue infections, Zika and AIUO, and dengue and AIUO at baseline, Days 3,7 and 28 of follow-up using Fisher's exact test and Wilcoxon rank-sum tests. The Holm procedure was used to control for multiple comparisons. There were 30 characteristics compared at baseline, so 90 comparisons were controlled for with the Holm procedure. There were 35 characteristics compared at days 3, so 105 comparisons were controlled for with the Holm procedure. There were 36 characteristics compared at day 7, so 108 comparisons were controlled for with the Holm procedure. There were 46 characteristics compared at day 28, so 138 comparisons were controlled for with the Holm procedure.

## Results

### Characteristics of the study population

We enrolled 467 patients with possible Zika infection, of which 26 were <12yo and excluded from this analysis. There were 266/441 (60%) women, and the median age of participants was 31yo (Range 12–76). Most patients (72%) had at least high-school education and lived in the city of Tapachula (73%). Among the 441 patients, 37 had Zika infection (8.4%), 73 dengue (16%) and 1 (0.2%) chikungunya infection. All others (n = 296, 67%) had neither of these and were classified as AIUO. We could not classify 34 (7.7%) events due to missing data. Table 1 describes the sociodemographic characteristics according to type of infection.

### Description of symptoms, physical exam and laboratory abnormalities at baseline

The median time between onset of any symptom and baseline visit was 4 days (range 0–7) but was shorter for patients with Zika (3 days, 0–6) than for patients with dengue (5 days, 0–7) (Table 1). While fever was, overall, the most frequent symptom (84%) it was self-reported less frequently by patients with Zika than by people with dengue (70.3% *vs*. 94.5%, *p* = 0.0611) (Table 2). Also, the length of fever before the first visit was shorter for patients with Zika than for dengue (3 days, range 0 to 6 *vs*. 5 days, range 1 to 7) (Table 1).

The distribution and type of self-reported rash was different between patients with dengue and all other groups (Table 2). While the frequency of self-reported rash was similar across all groups, the bodily distribution of the rash differs by group: 90.9% of patients with Zika

**Table 1. Baseline sociodemographic characteristics of 12 years and older patients seeking care within 7 days of onset of symptoms due to acute episodes of fever and/or rash (N = 441).**

| Characteristics | Confirmed Zika (n = 37) | Confirmed Dengue (n = 73) | Acute Illnesses of Unidentified Origin (n = 296) | Incomplete Data (n = 34) |
|---|---|---|---|---|
| Female | 23 (62.2%) | 36 (49.3%) | 187 (63.2%) | 20 (58.8%) |
| Age in years[1] | 33 (13, 59) | 27 (12, 68) | 32 (12, 76) | 28 (12, 49) |
| Ethnicity | | | | |
| White | 7 (18.9%) | 19 (26%) | 77 (26%) | 10 (29.4%) |
| Indigenous | 0 (0%) | 1 (1.4%) | 1 (0.3%) | 0 (0%) |
| Mestizo | 30 (81.1%) | 53 (72.6%) | 217 (73.3%) | 24 (70.6%) |
| Other | 0 (0%) | 0 (0%) | 1 (0.3%) | 0 (0%) |
| Residence | | | | |
| Tapachula | 20 (54.1%) | 42 (57.5%) | 232 (78.4%) | 28 (82.4%) |
| Other | 17 (45.9%) | 31 (42.5%) | 64 (21.6%) | 6 (17.6%) |
| Days between symptoms onset and enrollment[1] | | | | |
| Overall | 3 (0–6) | 5 (0–7) | 4 (0–7) | - |
| Fever | 3.0 (0–6) (n = 26) | 5.0 (1–7) (n = 69) | 3.0 (0–7) (n = 247) | - |
| Conjunctivitis | 2 (0–5) (n = 17) | 4 (1–7) (n = 24) | 3 (0–7) (n = 117) | - |

[1] median (range)

reported localized rash as opposed to patients with dengue that reported predominantly generalized rash (72.5%). Patients with AIUO with rash, reported it as generalized in 26.4% of cases. These differences were statistically significant (Table 2). Additionally, the frequency of type of rash differs by group: 50% of patients with Zika reported macular rash in comparison with 12.5% with dengue and 25.5% with AIUO.

**Table 2. Distribution and characteristics of baseline, self-reported signs and symptoms of patients 12 years and older seeking care within 7 days of onset due to acute episodes of fever and/or rash (N = 406).**

| Symptoms | Confirmed Zika Infection (n = 37) | Confirmed Dengue Infection (n = 73) | Acute Illnesses of Unidentified Origin (n = 296) | p-value[2] ZIKA vs. DENGUE | p-value[2] ZIKA vs. AIUO | p-value[2] DENGUE vs. AIUO |
|---|---|---|---|---|---|---|
| Fever (>37.2° C) | 26 (70.3%) | 69 (94.5%) | 248 (83.8%) | 0.0611 (0.0009) | 1.0000 (0.0643) | 0.8121 (0.0150) |
| Rash (self-reported) | 22 (59.5%) | 40 (54.8%) | 106 (35.8%) | 1.0000 (0.6878) | 0.3877 (0.0069) | 0.2908 (0.0048) |
| Initial location of rash | | | | | | |
| Generalized | 2 (9.1%) | 29 (72.5%) | 28 (26.4%) | **0.0002** (<0.0001) | 1.0000 (0.1003) | **0.00007** (<0.0001) |
| Localized | 20 (90.9%) | 11 (27.5%) | 78 (73.6%) | | | |
| Initial type of rash | | | | | | |
| Macular | 11 (50.0%) | 5 (12.5%) | 27 (25.5%) | 0.2697 (0.0043) | 0.4739 (0.0086) | 1.0000 (0.0306) |
| Other | 9 (40.9%) | 31 (77.5%) | 77 (72.6%) | | | |
| Petechial | 2 (9.1%) | 4 (10%) | 2 (1.9%) | | | |
| Arthralgia | 19 (51.4%) | 62 (84.9%) | 244 (82.4%) | **0.0306** (0.0004) | **0.0045** (<0.0001) | 1.0000 (0.7291) |
| Myalgia | 28 (75.7%) | 61 (83.6%) | 243 (82.1%) | 1.0000 (0.3194) | 1.0000 (0.3704) | 1.0000 (0.8647) |
| Conjunctivitis[1] | 17 (45.9%) | 26 (35.6%) | 123 (41.6%) | 1.0000 (0.3091) | 1.0000 (0.6023) | 1.0000 (0.4245) |
| Headache | 28 (75.7%) | 71 (97.3%) | 249 (84.1%) | 0.0572 (0.0008) | 1.0000 (0.2408) | 0.1105 (0.0017) |
| Malaise | 28 (75.7%) | 70 (95.9%) | 272 (91.9%) | 0.1586 (0.0025) | 0.3175 (0.0053) | 1.0000 (0.3195) |

[1] There was missing data about days of onset of conjunctivitis in 2 patients with dengue and 6 patients with undefined fever episodes.

[2] P-values are presented as adjusted an (unadjusted).

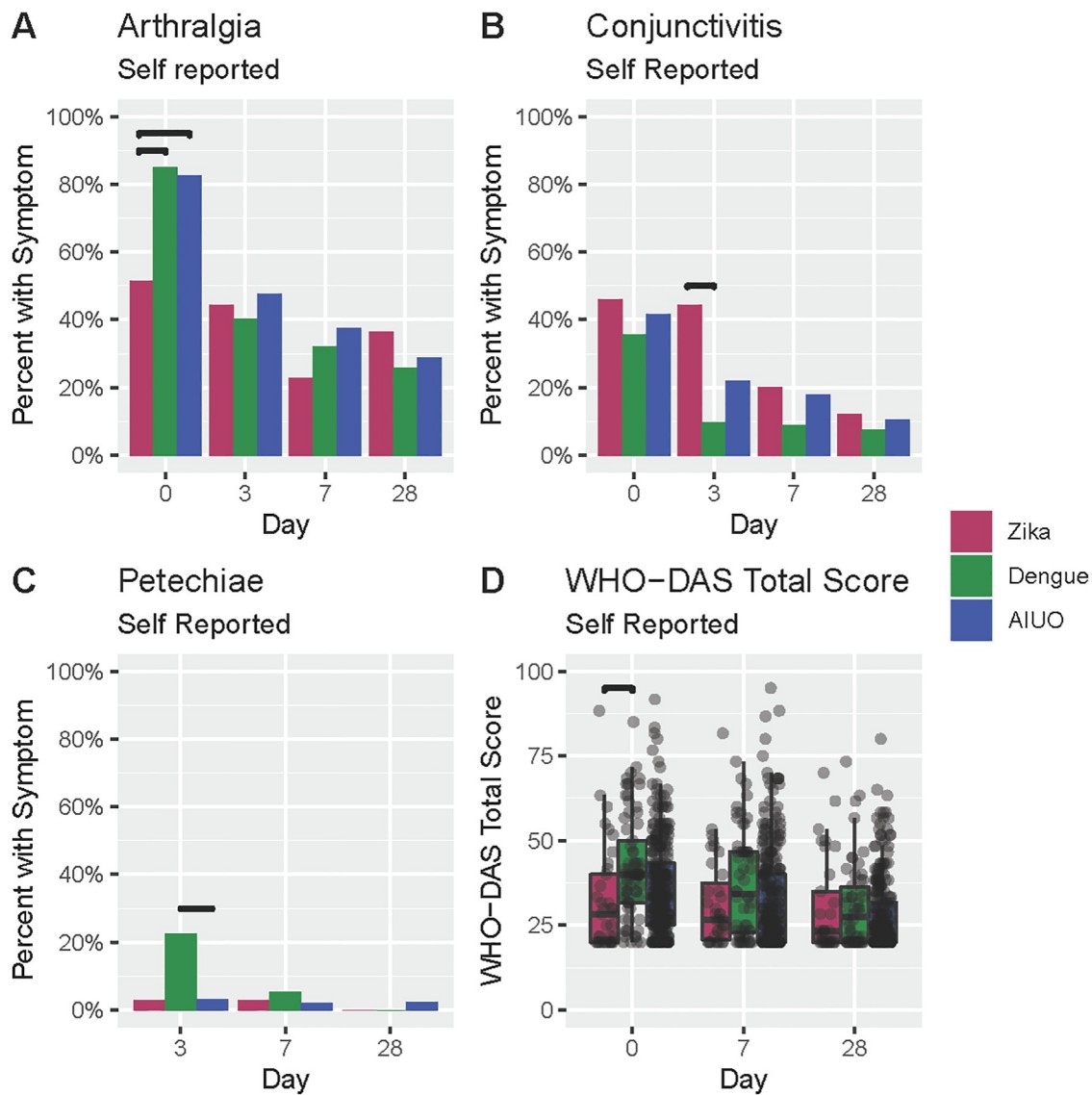

**Fig 1. Comparison of self-reported signs and symptoms among patients with confirmed Zika, dengue and AIUO.** Darker lines above bars indicate statistically significant comparisons.

After adjusting for multiple comparisons, these differences are not statistically significant. Notably, we observed no differences in the frequency of petechial rash between patients with Zika and dengue (9.1% and 10%) (Table 2). Physical exam corroborated the information on self-reported rash by patients in the initial visit, as described below.

We also observed noteworthy differences in the frequency of arthralgia (51.4% in Zika and 84.9% in dengue, $p = 0.0306$) (Fig 1A and Table 2), and headache (75.7% in Zika $vs$ 97.3% in dengue, $p = 0.0572$), but not in malaise (75.7% in Zika $vs.$ 95.9% in dengue, $p = 0.1586$), myalgia (75.7% in Zika $vs.$ 83.6% in dengue, $p = 0.03194$); nor in conjunctivitis (45.9% in Zika $vs$ 35.6% in dengue p = 1.0) (Fig 1B and Table 2).

At physical exam, rash was observed more frequently in patients with Zika and dengue when compared with those with AIUO (Table 3 and Fig 2A). A maculopapular rash was observed more frequently in patients with Zika (57.7%) than in dengue (11.5%) and AIUO

**Table 3. Baseline distribution and characteristics of physical exam of patients 12 years and older seeking care within 7 days of onset due to acute episodes of fever and/or rash (N = 406).**

| Physical exam | Zika Infection (n = 37) | Dengue Infection (n = 73) | Acute Illnesses of Unidentified Origin (n = 296) | p-value ZIKA vs DENGUE | p-value ZIKA vs AIUO | p-value DENGUE vs AIUO |
|---|---|---|---|---|---|---|
| **Rash at physical exam** | 26 (70.3%) | 52 (71.2%) | 121 (40.9%) | 1.0000 (1.0000) | 0.0553 (0.0008) | **0.0003** (<0.0001) |
| Maculopapular | 15 (57.7%) | 6 (11.5%) | 32 (26.4%) | **0.0024** (<0.0001) | 0.2777 (0.0045) | 1.0000 (0.0439) |
| Petechial | 2 (7.7%) | 10 (19.2%) | 8 (6.6%) | 1.0000 (0.3181) | 1.0000 (0.6907) | 1.0000 (0.0264) |
| Erythematous | 9 (34.6%) | 40 (76.9%) | 85 (70.2%) | **0.0314** (0.0004) | 0.0816 (0.0012) | 1.0000 (0.4597) |
| Other—combined with bruising | 1 (3.8%) | 2 (3.8%) | 3 (2.5%) | 1.0000 (1.0000) | 1.0000 (0.5478) | 1.0000 (0.6389) |
| **Injected conjunctivae** | 15 (40.5%) | 24 (32.9%) | 125 (42.2%) | 1.0000 (0.5274) | 1.0000 (1.0000) | 1.0000 (0.1827) |
| **Uveitis** | 5 (13.5%) | 4 (5.5%) | 38 (12.8%) | 1.0000 (0.1607) | 1.0000 (0.8004) | 1.0000 (0.0984) |
| **Petechiae at physical exam** | 1 (2.7%) | 17 (23.3%) | 13 (4.4%) | 0.3210 (0.0054) | 1.0000 (1.0000) | **0.0002** (<0.0001) |
| **Lymphadenopathy** | 16 (43.2%) | 33 (45.2%) | 123 (41.6%) | 1.0000 (1.0000) | 1.0000 (0.8611) | 1.0000 (0.5983) |
| **Any neurological abnormal finding** | 5 (13.9%)[1] | 26 (35.6%) | 81 (27.4%) | 1.0000 (0.0234) | 1.0000 (0.1060) | 1.0000 (0.1947) |

[1]One subject with missing data

(26.4%) (Fig 2B and Table 3) as opposed to the higher frequency of erythematous (76.9%) and petechial (19.2%) rash in patients with dengue (Fig 2C and Table 3). Petechia (Fig 2D and Table 3) was also observed more frequently in patients with dengue (23.3% *vs.* 2.7% in Zika and 4.4% in AIUO (Fig 1D and Table 3). The presence of any neurological abnormality during physical exam was observed more frequently in patients with dengue (35.6%) than Zika (13.9%) (Table 3).

Patients with dengue had the median highest score for disability at baseline (median score: 40, range: 20, 85). The differences in median WHODAS score at baseline were statistically significant when compared with patients with Zika (28.33, range: 20, 88.33, *p* = 0.0171) but not AIUO (33.33 (range: 20, 91.67, *p* = 0.0766) (Fig 1D and S1 Table).

There were also significant differences in laboratory values at baseline across groups. Overall, patients with dengue more frequently had thrombocytopenia (Fig 3A and S2 Table), leucopenia (Fig 3B and S2 Table), neutropenia (Fig 3C and S2 Table) lymphocytosis (Fig 3D and S2 Table), and higher alanine aminotransferase (ALT) (Fig 3F and S2 Table) than patients with Zika and AIUO. Nonetheless, patients with AIUO had higher C-Reactive Protein (CRP) and Erythrocyte Sedimentation Rate (ESR) (Fig 3E and S2 Table).

## Duration of clinical findings

On Day 3 visit only a few differences across groups remained, while other differences not seen at the baseline visit were observed. Overall, patients with Zika and dengue reported rash (55.6% and 56.5%) more frequently than patients with AIUO (27%). The frequency of conjunctivitis persisted in patients with Zika but decreased in the other two groups. Self-reported petechia was observed more frequent in patients with dengue (Fig 1C and S3 Table). We observed no clinically or statistically significant differences in the frequency of all other symptoms across groups (S3 Table).

In the physical exam at Day 3 visit (S4 Table), erythematous and petechial rash, were more commonly found among participants with confirmed dengue infection, while maculopapular rash was more frequently observed among participants with Zika, however these differences were not statistically significant at this point (Fig 2B and S4 Table). Injected conjunctivae were also less frequent among participants with dengue than in participants with Zika and AIUO.

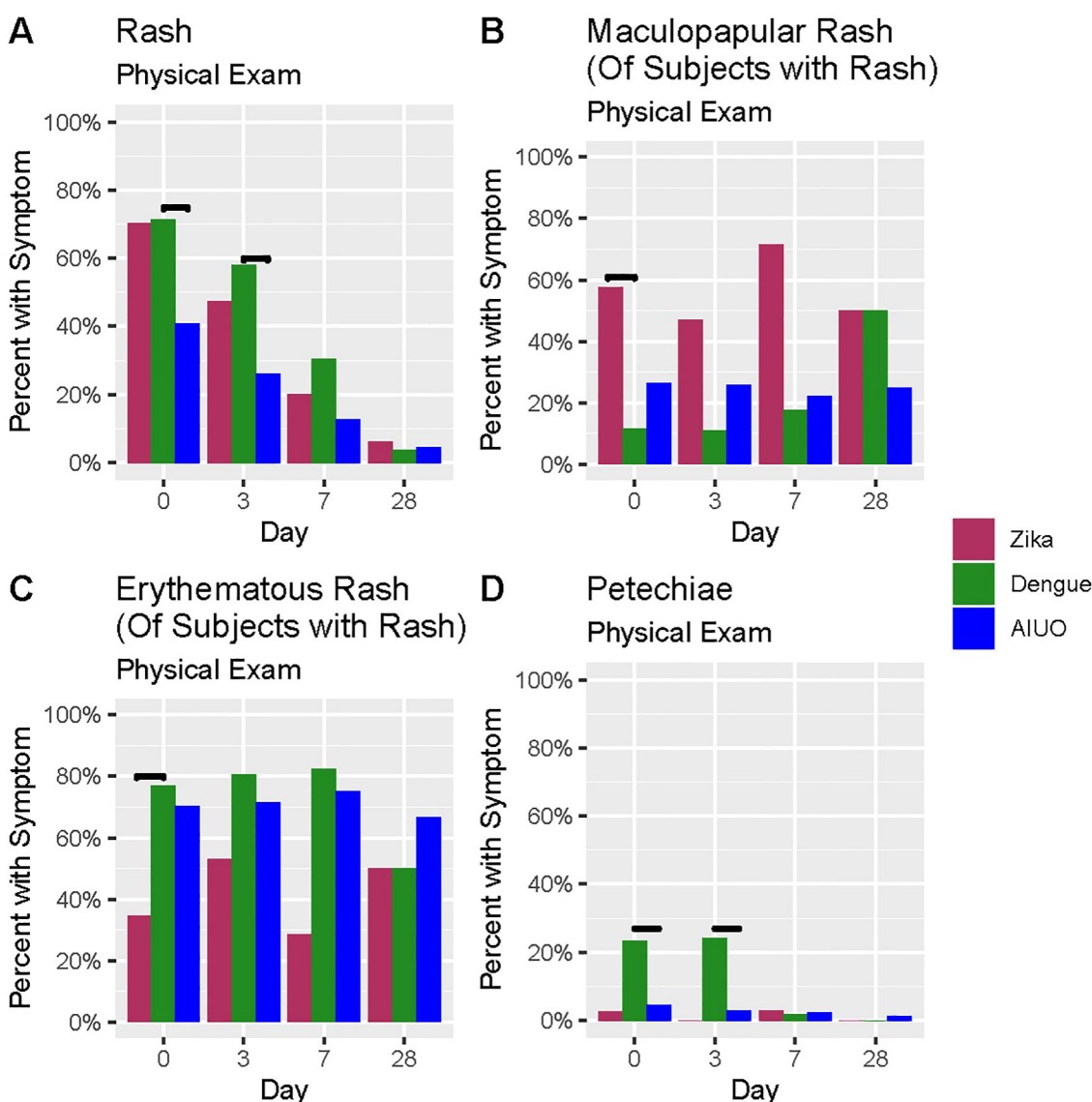

**Fig 2. Comparison of findings at physical exam among patients with confirmed Zika, dengue infections, and AIUO.** Darker lines above bars indicate statistically significant comparisons.

Among participants with AIUO, erythematous rash and injected conjunctivae were the most frequent clinical findings (S4 Table). None of these were statistically significant at Day 3 visit.

More patients with Zika and dengue had persistent self-reported rash than those with AIUO (28.6% and 30.4% *vs.* 13.7%) in the Day 7 visit, and a similar tendency was observed for itchiness (40% in Zika, 41.1% in dengue and 27.7% in AIUO) (S5 Table). Malaise, muscular weakness, fatigue, confusion or disorientation, and difficulty walking or standing upright were reported more frequently in people with dengue and AIUO compared to Zika. Sore throat, mouth ulcers and cough were reported more frequently by patients with AIUO than the other two groups (S5 Table). None of these differences were statistically significant at the Day 7 visit. Rash and neurological abnormalities were also observed more frequently at the Day 7 visit in patients with dengue (30.4% and 12.5%) when compared with Zika (20% and 2.9%) and AIUO (12.6% and 14.4%). Erythematous rash was more frequent in patients with dengue (82.4%),

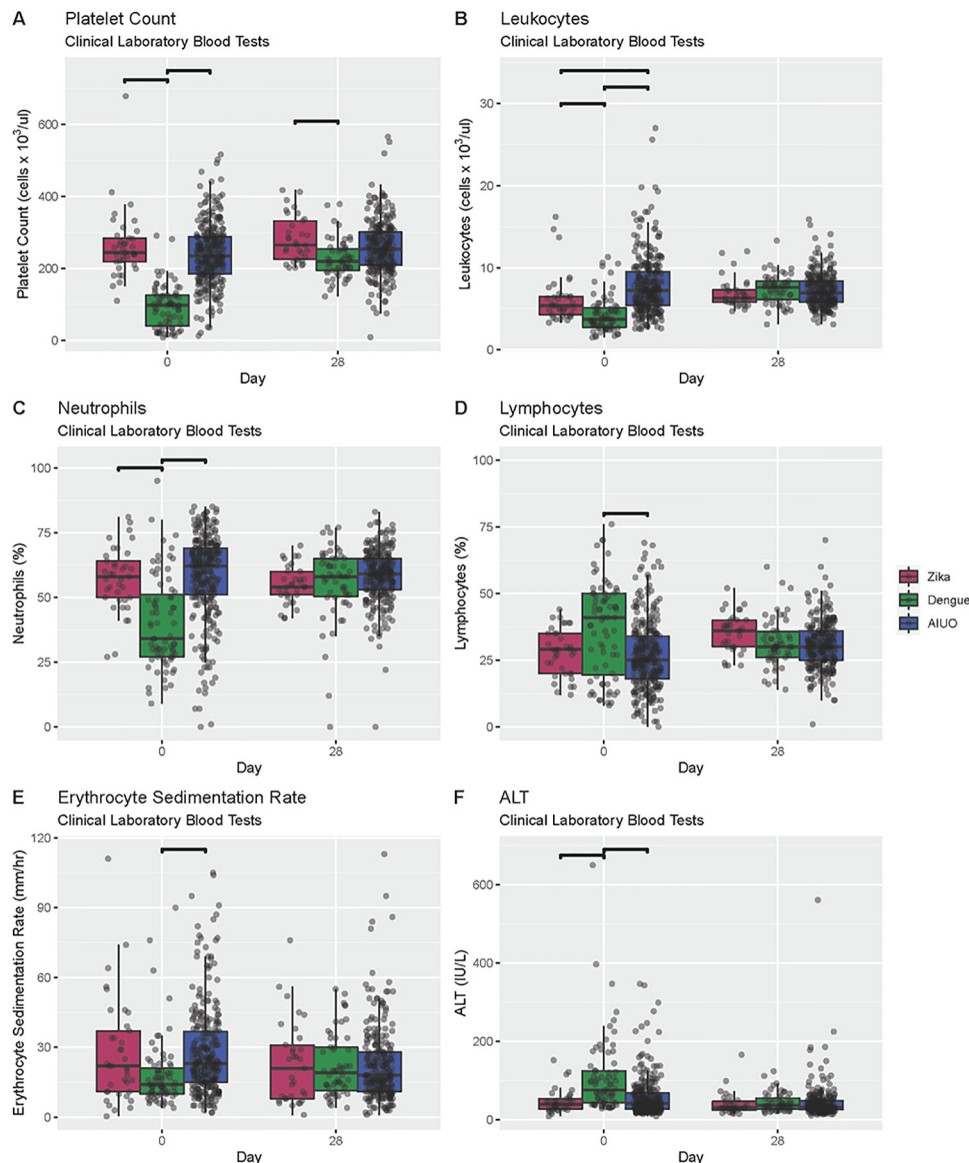

**Fig 3. Comparison of clinical laboratory blood tests among patients with confirmed Zika, dengue infections, and AIUO.** Darker lines above bars indicate statistically significant comparisons.

while maculopapular rash predominated in Zika (71.4%) at the Day 7 visit (S6 Table and Fig 2). Again, none of these differences were statistically significant.

There were no differences in the frequency of signs and symptoms at Day 28 visit across groups (S7 Table). Similarly, no differences were observed in physical exam nor in the laboratory blood tests across groups in this visit (S8 Table) exempt for the platelet count that was significantly lower in dengue patients compared with those with Zika (S9 Table and Fig 3A). We observed no persistence in the differences in the frequency of abnormal neurological findings after baseline visit (S7 Table). WHODAS median score were higher for the dengue group at Days 7 and 28 visits, but these differences were not statistically significant. (Fig 1D and S1 Table).

## Discussion

In this observational cohort study to characterize the natural history of Zika and compare clinical manifestations with dengue, chikungunya and acute illness of unidentified origin (AIUO); we identified that most episodes (67%) consistent with the PAHO definition of probable Zika case cannot be attributed to any flavivirus infection in this hyperendemic dengue area in the Mexico-Guatemala border. Zika accounted for 8% of episodes, dengue for 16%, and there was only 1 case (0.2%) of chikungunya infection. Our results show apparently distinctive but overlapping clinical patterns associated with Zika and dengue infections during the first 7 days of clinical manifestations, but most of these differences did not persist beyond 10 days after symptom initiation. Typically, dengue patients presented with fever, generalized non-macular rash, arthralgia, and petechiae more frequently than patients with Zika, resulting in a more debilitating disease. They also presented with laboratory abnormalities such as lower neutrophils and higher lymphocyte count, and abnormal liver function tests more frequently; which points to more severe disease with systemic manifestations. Conjunctivitis and localized macular rash were more frequent in patients with Zika. Notably, only conjunctivitis in patients with Zika persisted up to 10 days after symptom initiation. In contrast, patients with AIUO did not have a clinical finding that presented more frequently but this group had higher median leukocyte counts, neutrophilia and higher ESR. Self-reported rash and petechia in patients with dengue lasted longer when compared with AIUO. None of these differences persisted longer than 14 days after symptom initiation, consistent with rapid resolution of even the most severe symptoms. Only thrombocytopenia persisted up to Day 28 visit in patients with dengue.

Our findings are similar to previous studies in the frequency of clinical manifestations for Zika and dengue. For example, it has been previously noticed that people with Zika infection experience fever for shorter periods of time than patients with dengue [16]. We also observed that localized maculopapular rash is the most frequent type in patients with Zika, as opposed with the generalized, erythematous rash in dengue; confirming the results of a systematic review of 66 Zika case reports describing maculopapular rash as the more prevalent type [17]. While a third of patients with dengue were identified as having peripheral nervous system abnormalities at enrollment, three days later this decreased to 10%, which is consistent with a previous report in hospitalized patients with dengue in India [18]. Previously, pruritus and lymphadenopathies has been described in Zika infection, however, we observed that it is as frequent as in patients with dengue, and very common in other acute illnesses. The discrepancies with previous studies could be within the expected heterogeneity of clinical manifestations of the same disease [19].

Our study has the advantage of comparing systematically, prospectively, and simultaneously a substantial number of patients with different diseases that widely overlap geographically and clinically. Our data collection was preplanned with a structured visit schedule. This study documents a wider variety of clinical manifestations in patients with Zika, such as the relative frequency of peripheral neurological abnormalities at physical exam, which has been rarely explored before [20]. Our study, also simultaneously compares the duration of clinical manifestations in Zika with dengue and AIUO, showing that dengue tends to have a longer duration, except for conjunctivitis that persisted for longer in patients with Zika. In general, however, these differences in symptom duration did not persist longer than 10 days after symptoms initiation. Interestingly we observed that patients with dengue were more likely to be affected by peripheral neurological abnormalities than those with Zika. The only other parameters strongly associated with dengue and not with Zika and AIUO were thrombocytopenia and elevated levels of aspartate aminotransaminase, which is consistent with previous reports. Our results contribute to the growing body of evidence that

low platelet count might be a useful parameter to clinically distinguish Zika from dengue and other AIUO [5].

We identify several limitations in our study. First, we experienced a considerable loss to follow-up that was different across groups: only 82% of enrolled participants attended the Day 28 visit, which may have skewed the results of symptoms duration if, for example, the severity of specific symptoms was associated with attrition. We attempted to minimize some types of biases by standardizing data collection and performing physical examination and laboratory tests at scheduled visits. We also note that some of our data is self-reported asking patient to remember prior symptoms and there can be recall biases in this type of data. Nonetheless, we found good agreement between measurements of the same symptom by self-report and physical exam when examined at the same time. Misclassification bias due to erroneous attribution to disease group is always possible, as patients enrolled at different days after symptom initiation and viremia changes through the progression of the disease, potentially modifying sensitivity of PCR tests [21]. This error was minimized by repeated testing (increased specificity) simultaneously in different clinical samples (increased sensitivity).

In conclusion, we observed differences in patterns of clinical manifestations between patients with Zika, dengue and AIUO at the population level but it remains unclear whether symptoms alone can distinguish these diseases in individual patients given substantial overlap. The presence of some clinical manifestations such as thrombocytopenia can be a trigger for etiologic testing confirmation. The large proportion of patients with illness but AIUO shows the need for better diagnostic tools. Our study contributes to the literature on characterization of the variety and duration of clinical characteristics in patients with Zika, dengue and other acute illnesses. This is important since studies of Zika virus infections have small sample sizes and more reports are needed to fully understand the clinical effects of Zika virus.

## Supporting information

**S1 Table. Description of disability assessment during the first month of follow up using the WHODAS tool for disability among 441 patients 12 years and older seeking care within 7 days of onset due to acute episodes of fever and/or rash according to type infection.**
(PDF)

**S2 Table. Clinical laboratory blood tests at baseline visit of patients 12 years and older seeking care within 7 days of onset due to acute episodes of fever and/or rash (N = 406).**
(PDF)

**S3 Table. Distribution and characteristics of self-reported signs and symptoms at day 3 after baseline visit of patients 12 years and older seeking care within 7 days of onset due to acute episodes of fever and/or (N = 383).**
(PDF)

**S4 Table. Distribution and characteristics of physical exam 3 days after the first visit of patients 12 years and older seeking care within 7 days of onset due to acute episodes of fever and/or rash (N = 383).**
(PDF)

**S5 Table. Distribution and characteristics of self-reported signs and symptoms at day 7 after baseline visit of patients 12 years and older seeking care within 7 days of onset due to acute episodes of fever and/or rash (N = 376).**
(PDF)

**S6 Table. Distribution and characteristics of physical exam 7 days after the first visit of patients 12 years and older seeking care within 7 days of onset due to acute episodes of fever and/or rash (N = 376).**
(PDF)

**S7 Table. Distribution and characteristics of self-reported signs and symptoms at day 28 after baseline visit of patients 12 years and older seeking care within 7 days of onset due to acute episodes of fever and/or rash (N = 352).**
(PDF)

**S8 Table. Distribution and characteristics of physical exam 28 days after the first visit of patients 12 years and older seeking care within 7 days of onset due to acute episodes of fever and/or rash (N = 352).**
(PDF)

**S9 Table. Clinical laboratory blood tests at Day 28 visit of patients 12 years and older seeking care within 7 days of onset due to acute episodes of fever and/or rash (N = 352).**
(PDF)

## Acknowledgments

The members for the Mexico Emerging Infectious Diseases Clinical Research Network (LaRed) are Justino Regalado Pineda (LaRed Director), Héctor Armando Rincón-León and Karla R Navarro-Fuentes (Unidad de Medicina Familiar No.11, Instituto Mexicano del Seguro Social, Tapachula, Chiapas, Mexico), Sandra Caballero-Sosa, Francisco Camas-Durán and Zoyla Priego-Smith (Clínica Hospital Dr. Roberto Nettel Flores, Instituto de Seguridad y Servicios Sociales de los Trabajadores del Estado, Tapachula, Chiapas, Mexico), Emilia Ruiz (Hospital General de Tapachula, Tapachula, Chiapas, Mexico), José Gabriel Nájera-Cancino, Paul Rodriguez de La Rosa, Jesús Sepúlveda-Delgado, Alfredo Vera Maloof, Karina Trujillo, Alexander López-Roblero (Hospital Regional de Alta Especialidad Ciudad Salud, Tapachula, Chiapas, Mexico), Raydel Valdés-Salgado, Yolanda Bertucci, Isabel Trejos, Luis Diego Villalobos (Westat, Inc., Rockville, MD, USA), Pablo F Belaunzarán-Zamudio, Pilar Ramos, Fernando J. Arteaga-Cabello, Lourdes Guerrero, Guillermo Ruiz-Palacios (Departamento de Infectología, Instituto Nacional de Ciencias Médicas y Nutrición Salvador Zubirán, Mexico City, Mexico), Luis Mendoza-Garcés, Peter Quidgley, Eli Becerril, Paola del Carmen Guerra de Blas, Abelardo Montenegro Liendo (LaRed Coordinating Center, Mexico City), John H Powers III (Leidos Biomedical Research, Inc., Frederick National Laboratory for Cancer Research, Frederick, Maryland, USA), John H Beigel, Sally Hunsberger (National Institute of Allergy and Infectious Diseases, Bethesda, MD, USA).

## Disclaimer

The content of this publication does not necessarily reflect the views or policies of the Department of Health and Human Services, or Westat, nor does mention of trade names, commercial products, or organizations imply endorsement by the US Government.

## Author Contributions

**Conceptualization:** Pablo F. Belaunzarán-Zamudio, Sally Hunsberger, John H. Beigel, Guillermo Ruiz-Palacios.

**Data curation:** Allyson Mateja, Raydel Valdés-Salgado.

**Formal analysis:** Allyson Mateja, Sally Hunsberger.

**Funding acquisition:** Sophia Siddiqui, John H. Beigel, Guillermo Ruiz-Palacios.

**Investigation:** Héctor A. Rincón-León, Karla Navarro-Fuentes, Emilia Ruiz-Hernández, Sandra Caballero-Sosa, Francisco Camas-Durán, Zoila Priego-Smith, José G Nájera-Cancino, Alexander López-Roblero, Karina del Carmen Trujillo-Murillo.

**Methodology:** Pablo F. Belaunzarán-Zamudio, Sally Hunsberger, John H. Beigel, Guillermo Ruiz-Palacios.

**Project administration:** Sophia Siddiqui, John H. Beigel, Guillermo Ruiz-Palacios.

**Resources:** Sophia Siddiqui, John H. Beigel, Guillermo Ruiz-Palacios.

**Software:** Allyson Mateja, Sally Hunsberger, Raydel Valdés-Salgado.

**Supervision:** Pablo F. Belaunzarán-Zamudio, Sophia Siddiqui, John H. Beigel, Guillermo Ruiz-Palacios.

**Validation:** Allyson Mateja, Sally Hunsberger, Raydel Valdés-Salgado.

**Visualization:** Pablo F. Belaunzarán-Zamudio, Allyson Mateja, Sally Hunsberger.

**Writing – original draft:** Pablo F. Belaunzarán-Zamudio, Paola del Carmen Guerra-de-Blas.

**Writing – review & editing:** Pablo F. Belaunzarán-Zamudio, Allyson Mateja, Paola del Carmen Guerra-de-Blas, Héctor A. Rincón-León, Karla Navarro-Fuentes, Emilia Ruiz-Hernández, Sandra Caballero-Sosa, Francisco Camas-Durán, Zoila Priego-Smith, José G Nájera-Cancino, Alexander López-Roblero, Karina del Carmen Trujillo-Murillo, John H. Powers, Sally Hunsberger, Sophia Siddiqui, John H. Beigel, Raydel Valdés-Salgado, Guillermo Ruiz-Palacios.

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
