## [Decision Letter · Decision Letter 0]

3 Dec 2020

Dear Dr. Belaunzarán Zamudio,

Thank you very much for submitting your manuscript "Comparison of clinical characteristics of Zika and dengue symptomatic infections and other acute illnesses of unidentified origin in Mexico" for consideration at PLOS Neglected Tropical Diseases. As with all papers reviewed by the journal, your manuscript was reviewed by members of the editorial board and by several independent reviewers. The reviewers appreciated the attention to an important topic. Based on the reviews, we are likely to accept this manuscript for publication, providing that you modify the manuscript according to the review recommendations. 

Sincerely,

David Joseph Diemert, M.D.

Associate Editor

Amy Morrison

Deputy Editor

Reviewer's Responses to Questions

**Key Review Criteria Required for Acceptance?**

**Methods**

-Are the objectives of the study clearly articulated with a clear testable hypothesis stated?

-Is the study design appropriate to address the stated objectives?

-Is the population clearly described and appropriate for the hypothesis being tested?

-Is the sample size sufficient to ensure adequate power to address the hypothesis being tested?

-Were correct statistical analysis used to support conclusions?

-Are there concerns about ethical or regulatory requirements being met?

Reviewer #1: Yes the objective of this study was to provide a detailed clinical description of zika/dengue/chikungunya

This is a an observational, prospective, cohort study which is appropriate to answer the objective of clinical description. 

Enrolled 467, surprising that 296 patients had unknown origin for their acute illness. For the period of time, the sample size is sufficient for a pilot study to lead to larger study. 

No concerns for ethical or regulatory

Reviewer #2: The objectives are clear and the design is appropriate.

**Results**

-Does the analysis presented match the analysis plan?

-Are the results clearly and completely presented?

-Are the figures (Tables, Images) of sufficient quality for clarity?

Reviewer #1: Yes the authors fulfill these criteria. 

The figures are easy to read and follow and answer the clinical question of differentiating between the different arboviral infections.

Reviewer #2: The results are appropriate. Some repetition could be reduced and the supplemental tables need to be properly labeled with footnotes defined.

**Conclusions**

-Are the conclusions supported by the data presented?

-Are the limitations of analysis clearly described?

-Do the authors discuss how these data can be helpful to advance our understanding of the topic under study?

-Is public health relevance addressed?

Reviewer #1: The authors present their conclusion in a clear and concise way. They enumerated the different limitations of the study addressing loss to follow up, self-reporting, and misclassification bias.

Reviewer #2: The conclusions are supported by the data presented.

**Editorial and Data Presentation Modifications?**

Reviewer #1: -Background: remove "comma" after clinical description

-Punctuation marks need to be reviewed: missing commas or passive voice sentences

Reviewer #2: The paper would benefit from editing by a native English speaker.

**Summary and General Comments**

Reviewer #1: An interesting observational cohort study is trying to differentiate the three arboviral diseases clinically by looking at clinical and laboratory tests. As a clinician, the study illustrated the subtle difference between the three arboviral diseases, and this might be helpful in the clinical setting where there is no PCR. Interestingly, was the majority of patients had an unknown disease, which could be other vector-borne diseases. The study accomplishes this even though it is observational. This study can lead to a bigger prospective study to see if their findings are accurate in the diagnosis of zika, dengue, and chikungunya. The authors explain well their methodology and results. Conclusions are to the point and summarize the key points and, at the same time, point out the limitation of the study .

Reviewer #2: This is an important study comparing the clinical presentation of dengue, zika and fever of unknown origin. Because of lack of rapid diagnostic testing, defining clinical and laboratory differences between dengue and other causes of acute febrile illness (AFI) is important. The paper would benefit from being edited by a native English speaker. In the abstract, laboratory diagnosis needs to be described. It is not clear why 8.6% of confirmed dengue and zika cases were excluded, please clarify. Other AFI are not included in the abstract and could be removed from the methods. Clarify when the follow-up visits occurred. The abstract says 3, 7, and 28 days and the methods up to 6 months. Remove the 1 chikungunya case form table 1. Not much can be said from an n=1 and it is distracting. A brief description of what the disability assessment measures would be helpful. The differences in the frequency of rash are repeated in the results in two different places, I suggest you combined and discuss it in the same paragraph, otherwise it seems repetitive. The supplemental tables, need titles.

PLOS authors have the option to publish the peer review history of their article (what does this mean?). If published, this will include your full peer review and any attached files.

Reviewer #1: Yes: Laila Woc-Colburn

Reviewer #2: No
---

## [Editor Report · Decision Letter 1]

12 Jan 2021

Dear Dr. Belaunzarán Zamudio,

We are pleased to inform you that your manuscript 'Comparison of clinical characteristics of Zika and dengue symptomatic infections and other acute illnesses of unidentified origin in Mexico' has been provisionally accepted for publication in PLOS Neglected Tropical Diseases.

Best regards,

David Joseph Diemert, M.D.

Associate Editor

Amy Morrison

Deputy Editor

---

## [Editor Report · Acceptance letter]

10 Feb 2021

Dear Dr. Belaunzarán Zamudio,

We are delighted to inform you that your manuscript, "Comparison of clinical characteristics of Zika and dengue symptomatic infections and other acute illnesses of unidentified origin in Mexico," has been formally accepted for publication in PLOS Neglected Tropical Diseases.

Best regards,

Shaden Kamhawi

co-Editor-in-Chief

Paul Brindley

co-Editor-in-Chief
